# Methotrexate attenuates vascular inflammation through an adenosine-microRNA-dependent pathway

Dafeng Yang[1,2†], Stefan Haemmig[1†], Haoyang Zhou[1†], Daniel Pérez-Cremades[1], Xinghui Sun[1], Lei Chen[1,2], Jie Li[1], Jorge Haneo-Mejia[3,4], Tianlun Yang[5], Ivana Hollan[1,6,7], Mark W Feinberg[1]*

[1]Department of Medicine, Cardiovascular Division, Brigham and Women's Hospital, Harvard Medical School, Boston, United States; [2]Department of Cardiovascular Surgery, The Second Xiangya Hospital, Central South University, Changsha, China; [3]Department of Pathology and Laboratory Medicine, Institute for Immunology, Perelman School of Medicine, University of Pennsylvania, Philadelphia, United States; [4]Division of Protective Immunity, Department of Pathology and Laboratory Medicine, Children's Hospital of Philadelphia, Philadelphia, United States; [5]Department of Cardiology, Xiangya Hospital, Central South University, Changsha, China; [6]Lillehammer Hospital for Rheumatic diseases, Lillehammer, Norway; [7]Norwegian University of Science and Technology, Gjøvik, Norway

**Abstract** Endothelial cell (EC) activation is an early hallmark in the pathogenesis of chronic vascular diseases. MicroRNA-181b (*Mir181b*) is an important anti-inflammatory mediator in the vascular endothelium affecting endotoxemia, atherosclerosis, and insulin resistance. Herein, we identify that the drug methotrexate (MTX) and its downstream metabolite adenosine exert anti-inflammatory effects in the vascular endothelium by targeting and activating *Mir181b* expression. Both systemic and endothelial-specific *Mir181a2b2*-deficient mice develop vascular inflammation, white adipose tissue (WAT) inflammation, and insulin resistance in a diet-induced obesity model. Moreover, MTX attenuated diet-induced WAT inflammation, insulin resistance, and EC activation in a *Mir181a2b2*-dependent manner. Mechanistically, MTX attenuated cytokine-induced EC activation through a unique adenosine-adenosine receptor A3-SMAD3/4-*Mir181b* signaling cascade. These findings establish an essential role of endothelial *Mir181b* in controlling vascular inflammation and that restoring *Mir181b* in ECs by high-dose MTX or adenosine signaling may provide a potential therapeutic opportunity for anti-inflammatory therapy.

*For correspondence:
mfeinberg@bwh.harvard.edu

†These authors contributed equally to this work

Competing interests: The authors declare that no competing interests exist.

## Introduction

Activated endothelial cells (ECs) orchestrate the expression of adhesion molecules and release chemokines to foster the recruitment of leukocytes into the vessel wall. This presents an early hallmark that contributes to the development of chronic vascular disease states, such as atherosclerosis, insulin resistance, and rheumatoid arthritis (RA) (*Khan et al., 2010*; *Rao et al., 2007*; *Speyer and Ward, 2011*; *Sun et al., 2014a*; *MICU Registry et al., 2012*; *Sun et al., 2016*). Accumulating studies demonstrate that targeting EC activation and/or dysfunction may open new therapeutic approaches for the prevention and treatment of inflammatory diseases (*Chen et al., 2017*; *Gareus et al., 2008*; *Nus et al., 2016*). For example, inhibition of NF-κB activity specifically in the vascular endothelium significantly reduced atherosclerotic plaque formation and progression (*Gareus et al., 2008*).

MicroRNAs (miRNAs), a class of evolutionary conserved small non-coding RNAs, are important post-transcriptional regulators of gene expression through mRNA degradation or translational repression (*Loyer et al., 2014*; *Sun et al., 2013*). We have previously identified microRNA-181b (*Mir181b*), an anti-inflammatory miRNA, which prevents EC activation and diverse vascular inflammatory disease states including endotoxemia, atherosclerosis, insulin resistance, and arterial thrombosis (*Lin et al., 2016*; *Sun et al., 2014a*; *MICU Registry et al., 2012*; *Sun et al., 2016*), suggesting that *Mir181b* could serve as a promising target for preventing EC activation and dysfunction associated with inflammatory diseases. *Mir181b* expression is significantly reduced in response to a range of pro-inflammatory stimuli in the vascular endothelium (*Lin et al., 2016*; *Sun et al., 2014a*; *MICU Registry et al., 2012*; *Sun et al., 2016*). Therefore, identification of signaling pathways or therapeutics to rescue *Mir181b* expression in the vascular endothelium could open new strategies to control vascular inflammation.

Methotrexate (MTX) is a first-line treatment in RA and other autoimmune-mediated inflammatory diseases. Observational studies suggested that low-dose MTX could reduce cardiovascular risk in patients with RA (*Charles-Schoeman et al., 2017*; *Deyab et al., 2017*; *Hjeltnes et al., 2013*; *Kisiel et al., 2015*). For example, the Psoriatic arthritis, Ankylosing spondylitis, Rheumatoid Arthritis Study (PSARA) demonstrated that patients receiving MTX improved endothelial function and reduced E-selectin plasma levels after 6 months follow-up (*Deyab et al., 2017*; *Hjeltnes et al., 2013*). Another clinical trial identified that MTX treatment significantly lowered carotid intima-media thickness in RA patients compared to placebo (*Kisiel et al., 2015*). However, the recent Cardiovascular Inflammation Reduction Trial (CIRT), designed to determine the effect of low-dose MTX on cardiovascular secondary prevention, found that low-dose MTX did not reduce cardiovascular events compared to placebo (*Ridker et al., 2019*). These conflicting results from different clinical trials raise questions for the mechanisms of MTX-mediated effects on vascular inflammation. While adenosine has been identified as a potential metabolite activated in response to MTX in ECs (*Haskó and Cronstein, 2013*), gaps remain in our understanding of the downstream signaling pathways underlying its effects.

In this study, we identify that high-dose MTX and its downstream metabolite adenosine exert anti-inflammatory effects in the vascular endothelium through an MTX-adenosine receptor A3-*Mir181b*-dependent signaling pathway. These results provide a general paradigm for therapeutic control of miRNA expression and that restoring *Mir181b* expression in the vascular endothelium by high-dose MTX or adenosine signaling may provide therapeutic opportunities for anti-inflammatory therapy.

## Results

### MTX attenuates TNF-α-induced EC activation by upregulation of MIR181B-2 expression

Because of their known anti-inflammatory effects in the vascular endothelium (*Diamantis et al., 2017*; *Mangoni et al., 2017*), we explored whether MTX or HMG-CoA reductase inhibitors (statins) activated the expression of the anti-inflammatory *MIR181B* (*Lin et al., 2016*; *Sun et al., 2014a*; *MICU Registry et al., 2012*; *Sun et al., 2016*). MTX (10 μM) but not statins significantly upregulated *MIR181B* expression by 1.35-fold in cultured human umbilical vein endothelial cells (HUVECs) (*Figure 1A*). The MTX-induced expression of *MIR181B* was dose- and time-dependent with the highest induction at 10 μM and 4 hr (*Figure 1B,C*). Previous studies have established that adenosine (Ad) is the major anti-inflammatory effector of MTX, which is a product of adenosine monophosphate, a reaction catalyzed by the enzyme ecto-5' nucleotidase (*Chan and Cronstein, 2013*; *Gadangi et al., 1996*; *Tian and Cronstein, 2007*). When using a competitive enzyme inhibitor of ecto-5' nucleotidase (α, β-methylene adenosine-5'-diphosphate (APCP)), the MTX-induced *MIR181B* expression was completely blocked (*Figure 1D*). Conversely, Ad alone increased *MIR181B* expression in a dose- and time-dependent manner with the highest observed increase by 2.3-fold at 50 μM and 4 hr (*Figure 1E,F*).

*MIR181B* expression is rapidly reduced in response to pro-inflammatory stimuli such as TNF-α in ECs and overexpression *MIR181B* suppressed TNF-α-induced EC activation by repressing *VCAM-1*, *E-selectin*, and *ICAM-1* expression (*MICU Registry et al., 2012*; *Yamasaki et al., 2003*). To address

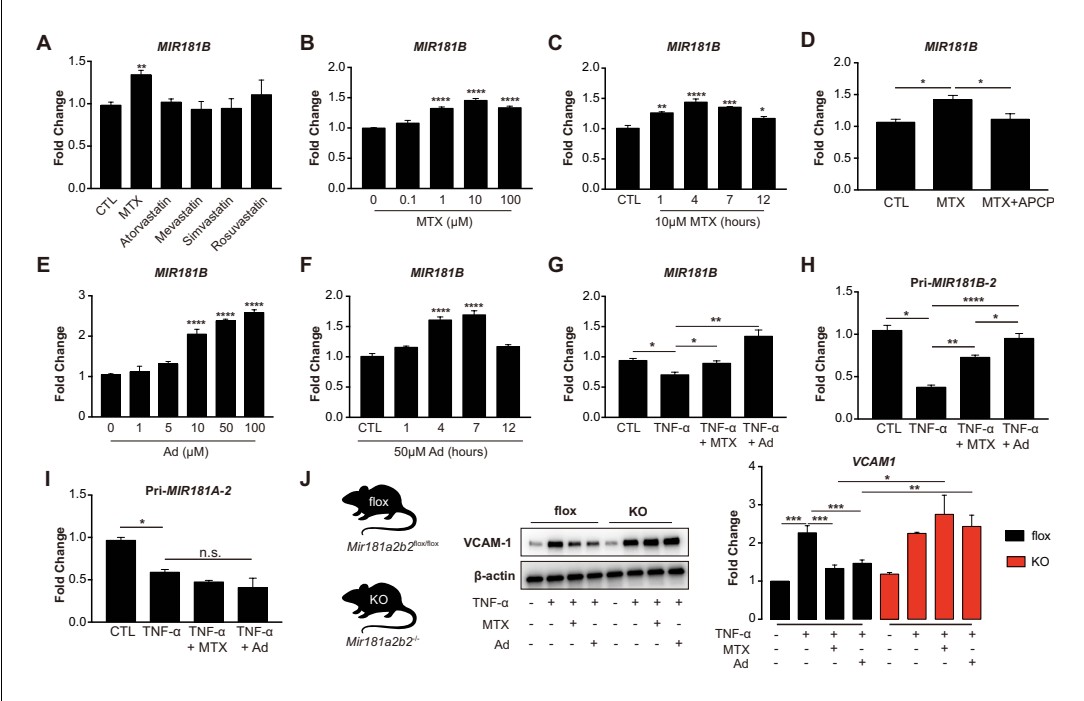

**Figure 1.** Methotrexate (MTX) represses TNF-α-induced pro-inflammatory gene expression via upregulation of *MIR181B-2* expression in ECs. (**A**) Real-time qPCR analysis of mature *MIR181B* in HUVECs in the presence or absence of MTX (10 μM), Atorvastatin (1 μM), Mevastatin (1 μM), Simvastatin (1 μM), and Rosuvastatin (1 μM) for 4 hr. three biological replicates. Unpaired two-tailed Student *t* test. (**B**) Titration of MTX (0 to 100 μM) for 4 hr (three biological replicates, Unpaired two-tailed Student *t* test) and (**C**) time course of MTX (10 μM) in HUVECs to assess *MIR181B* expression, three biological replicates. One-way ANOVA. (**D**) Real-time qPCR analysis of *MIR181B* in HUVECs incubated with MTX (10 μM) or in combination with APCP (50 μM) for 4 hr. Three biological replicates. Unpaired two-tailed Student *t* test. (**E**) Dose-response of adenosine (Ad) (0 to 100 μM) for 4 hr and (**F**) time course of Ad (50 μM) over 0–12 hr on *MIR181B* expression in HUVECs. Three biological replicates. One-way ANOVA. (**G**) HUVECs were treated with TNF-α (10 ng/ml) alone or in combination with either MTX (10 μM) or Ad (50 μM) for 4 hr. Three biological replicates. Unpaired two-tailed Student *t* test. Analysis of primary transcript of (**H**) *MIR181B-2* (*Pri-MIR181B-2*) or (**I**) *MIR181A-2* (*Pri-MIR181A-2*) in response to TNF-α (10 ng/ml) with or without MTX (10 μM) or Ad (50 μM) for 4 hr in HUVECs. Three biological replicates. Unpaired two-tailed Student *t* test. (**J**) Isolated primary lung endothelial cells (ECs) from *Mirr181a2b2*^flox/flox (flox) mice and *Mir181a2b2*^-/- (KO) mice were treated with TNF-α (10 ng/ml) with or without MTX (10 μM) or Ad (50 μM) for 8 hr to analyze *VCAM-1* protein expression. Please see *Figure 1—source data 1*. Three biological replicates. Unpaired two-tailed Student *t* test. *p<0.05; **p<0.01; ***p<0.001; ****p<0.0001. n.s. indicated non significance. All values represent mean ± SEM.
The online version of this article includes the following source data and figure supplement(s) for figure 1:

**Source data 1.**
**Figure supplement 1.** Methotrexate (MTX) and Ad repress TNF-α-induced pro-inflammatory genes without affecting primary *MIR181A-1* and *MIR181B-1* expression.

whether MTX or Ad may rescue the TNF-α-mediated reduction in *MIR181B* expression, HUVECs were stimulated with TNF-α in combination with either MTX or Ad. Treatment with MTX or Ad fully rescued the TNF-α repression of *MIR181B* expression (*Figure 1G*). As expected, ECs treated with MTX or Ad significantly reduced *VCAM-1*, *ICAM-1,* and *E-selectin* expression stimulated by TNF-α (*Figure 1—figure supplement 1A,B*). Because two loci for *MIR181B* (i.e. *MIR181B-1* and *MIR181B-2*) exist (*Sun et al., 2014b*), we examined which locus is regulated by TNF-α, MTX, or Ad. To this end, primary miRNA transcripts were analyzed in response to stimulation with TNF-α. Only primary transcript of the *MIR181B-2* locus, and not primary *MIR181B-1*, *MiR-181A-1*, or *MiR-181A-2*, was rescued by MTX or Ad co-stimulation (*Figure 1H,I*; *Figure 1—figure supplement 1C,D*).

To study the relationship between MTX or Ad to *Mir181b* in mouse tissues, primary lung ECs were isolated from *Mir181a2b2*^−/− knockout mice (KO), reflecting deficiency of *Mir181a-2* and *Mir181b-2*, or control *Mir181a2b2*^flox/flox (flox). Because *Mir181a-2* and *Mir181b-2* are located in close proximity to each other, the knockout strategy deleted both miRNAs (*Henao-Mejia et al., 2013*). Isolated primary ECs were stimulated with TNF-α in the presence or absence of MTX or Ad. In ECs isolated from wild-type flox mice, MTX or Ad inhibited VCAM-1 protein expression by 40%

and 35%, respectively. However, MTX or Ad did not reduce VCAM-1 expression in primary ECs from *Mir181a2b2*$^{-/-}$ KO mice (*Figure 1J*). Taken together, these findings suggest that pri-*Mir181b-2* and its mature isoform *Mir181b* is increased by MTX or Ad, and may mediate the protective anti-inflammatory effects of MTX or Ad on TNF-α-induced EC activation.

## Adenosine receptor A3 (ADORA3) mediates MTX-induced expression of MIR181B in ECs

Ad is released by cells to the extracellular environment at a low concertation and acts as a local modulator with a generally cytoprotective function through interaction with its four cell surface Ad receptors, *ADORA1*, *ADORA2A*, *ADORA2B,* and *ADORA3*. Each of these Ad receptors have a unique pharmacological profile, tissue distribution, and effector coupling (*Haskó and Cronstein, 2004*; *Jacobson and Gao, 2006*). To examine which Ad receptor mediates the MTX-induced *MIR181B* expression, individual knockdown for each of the Ad receptors was performed in ECs first in the absence of MTX or Ad (*Figure 2—figure supplement 1A–H*). Only silencing of *ADORA3* reduced *MIR181B* basal expression by 40%, while knockdown of *ADORA1*, *ADORA2A, and ADORA2B* had no effect on *MIR181B* expression (*Figure 2A*). This suggests that *ADORA3* might mediate the *MIR181B* expression activated by MTX or Ad. Therefore, we next examined whether AdoRA$_3$ knockdown blocked the MTX or Ad-mediated increase in *MIR181B* expression. Indeed, neither MTX nor Ad increased *MIR181B* expression in *ADORA3*-deficient HUVECs compared to negative control siRNA (*Figure 2B*). *MIR181B* expression is reduced in response to pro-inflammatory stimuli such as TNF-α (*Lin et al., 2016*; *Sun et al., 2014a*; *MICU Registry et al., 2012*; *Sun et al., 2016*). Treatment of ECs with TNF-α in the presence of MTX or Ad rescued the TNF-α-mediated repression of *MIR181B* expression in an AdoRA$_3$-specific manner (*Figure 2C* and *Figure 2—figure supplement 1A*). Furthermore, silencing of *ADORA3*, (and not *ADORA1*, *ADORA2A*, or *ADORA2B*) abrogated the anti-inflammatory effect of MTX or Ad in TNF-α-activated ECs on *VCAM-1*, *ICAM-1*, and *E-Selection* expression (*Figure 2D* and *Figure 2—figure supplement 1B*). Finally, to further assess whether the anti-inflammatory effect of MTX or Ad is mediated through an *ADORA3-MIR181B* signaling cascade, knockdown of *ADORA3* was combined with simultaneous overexpression of *MIR181B* mimic. While the MTX or Ad anti-inflammatory effects were blocked in the presence of *ADORA3* silencing, overexpression of *MIR181B* fully rescued the MTX and Ad-mediated reduction of ECs activation markers (*Figure 2E*). In summary, our data indicate that *ADOR3* plays central role in mediating MTX- or Ad-induced *MIR181B* expression, and this *ADORA3-MIR181B* signaling pathway contributes to the inhibitory effects of MTX or Ad on TNF-α-induced EC activation.

## MTX and Ad regulate MIR181B-2 promoter activity via SMAD3/4

To address, how MTX regulates *MIR181B-2* expression on the transcriptional level, different lengths of the proximal *MIR181A2B2* promoter (containing sequences 5' to both MIR181A-2 and MIR181B-2) were cloned upstream of a luciferase reporter. MTX and Ad activated the −606 bp *MIR181A2B2* promoter reporter construct from 1.47- to 1.57-fold, respectively (*Figure 3A–B*). Treatment of MTX or Ad similarly activated the −402 bp reporter comparable to the full-length constructs (*Figure 3B*). However, reducing the promoter length to −301 bp or −150 bp significantly reduced luciferase activity from 30- to 2600-fold, respectively (*Figure 3B*). At the post-transcriptional level, mRNA stability is an important factor determining mRNA abundance. To investigate whether MTX or Ad would affect pri-*MIR181B-1* or pri- *MIR181B-2* transcript stability, we used the transcription inhibitor actinomycin D in MTX or Ad-treated HUVECs. We found that neither MTX nor Ad affected the stability of pri-*MIR181B-1* transcript (*Figure 3—figure supplement 2A*). However, MTX significantly increased pri-*MIR181B-2* transcript stability, while Ad had a modest opposite effect (*Figure 3—figure supplement 2B*). These results indicate that the promoter region between −402 and −301 is likely required for MTX and Ad-induced transcriptional activity of the *MIR181A2B2* locus and MTX and Ad affects pri-*MIR181B-2* expression at both transcriptional and post-transcriptional levels. To further identify specific transcription factor binding sites, in silico prediction tools identified a SMAD2/3/4 binding site in the region −402 to −301 (*Figure 3B*). This was further investigated by individual knockdown of *SMAD2*, *SMAD3*, or *SMAD4* (*Figure 3—figure supplement 3*). Silencing of *SMAD3* or *SMAD4* reduced luciferase activity of the full length −606 bp construct by 37% and 42%, respectively, in ECs compared to control siRNA (*Figure 3C*). However, *SMAD2* silencing did not

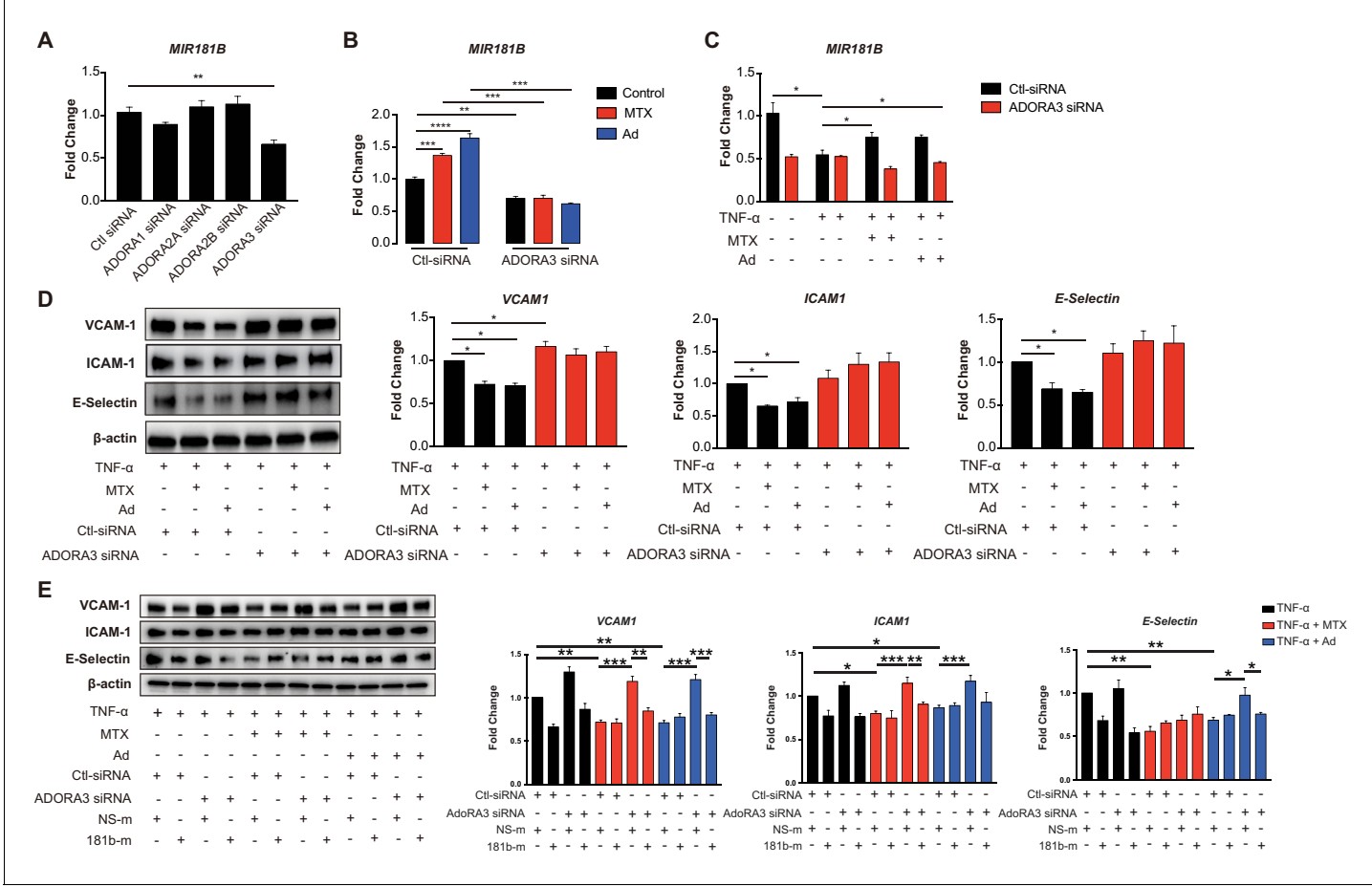

**Figure 2.** Induction of *MIR181B* expression by methotrexate (MTX) or adenosine is adenosine receptor A₃ (*ADORA3*) dependent. (**A**) Knockdown for adenosine receptors A1, A2A, A2B, and A3 in HUVECs was performed to analyze *MIR181B* expression. three biological replicates. One-way ANOVA. (**B**) *MIR181B* expression in HUVECs transfected with Ctl-siRNA or *ADORA3* siRNA after treatment with MTX (10 μM) or Ad (50 μM) or (**C**) treatment with TNF-α (10 ng/ml) alone or in combination MTX (10 μM) or Ad (50 μM). Three biological replicates. One-way ANOVA and Unpaired two-tailed Student *t* test. (**D**) Western blot analyses of *VCAM-1*, *ICAM-1*, and *E-Selectin* expression in HUVECs transfected with Ctl-siRNA or *ADORA3* siRNA in the presence of TNF-α (10 ng/ml) in combination with either MTX (10 μM) or Ad (50 μM). Three biological replicates. Unpaired two-tailed Student *t* test. (**E**) in the presence of miRNA negative control (NS-m) or *MIR181B* mimics (181b-m) stimulated with TNF-α (10 ng/ml) or in combination with MTX (10 μM) or Ad (50 μM). Please see *Figure 2—source data 1*. Three biological replicates. Unpaired two-tailed Student *t* test. *p<0.05; **p<0.01; ***p<0.001; ****p<0.0001. n.s. indicated non significance. All values represent mean ± SEM.

The online version of this article includes the following source data and figure supplement(s) for figure 2:

**Source data 1.**

**Figure supplement 1.** Knockdown efficiency for adenosine receptor siRNAs.

impact luciferase activity (*Figure 3C*). Consistent with this observation, *MIR181B* expression was significantly reduced only by silencing *SMAD3* or *SMAD4*, and not by *SMAD2* knockdown (*Figure 3D*). Importantly, silencing of *SMAD3* or *SMAD4* completely blocked the MTX and Ad-mediated increase in *MIR181B* expression (*Figure 3E*). Furthermore, knockdown *ADORA3* blocked the *MIR181A2B2* promoter luciferase activity induced by MTX and Ad (*Figure 3F*); however, silencing of the other three Ad receptors had no effect on luciferase activity (*Figure 3G*). Taken together, these data indicate that *SMAD3* and *SMAD4* are not only required for transcriptional activity of *MIR181B* expression, but also significantly contribute to MTX and Ad-induced *MIR181B* expression.

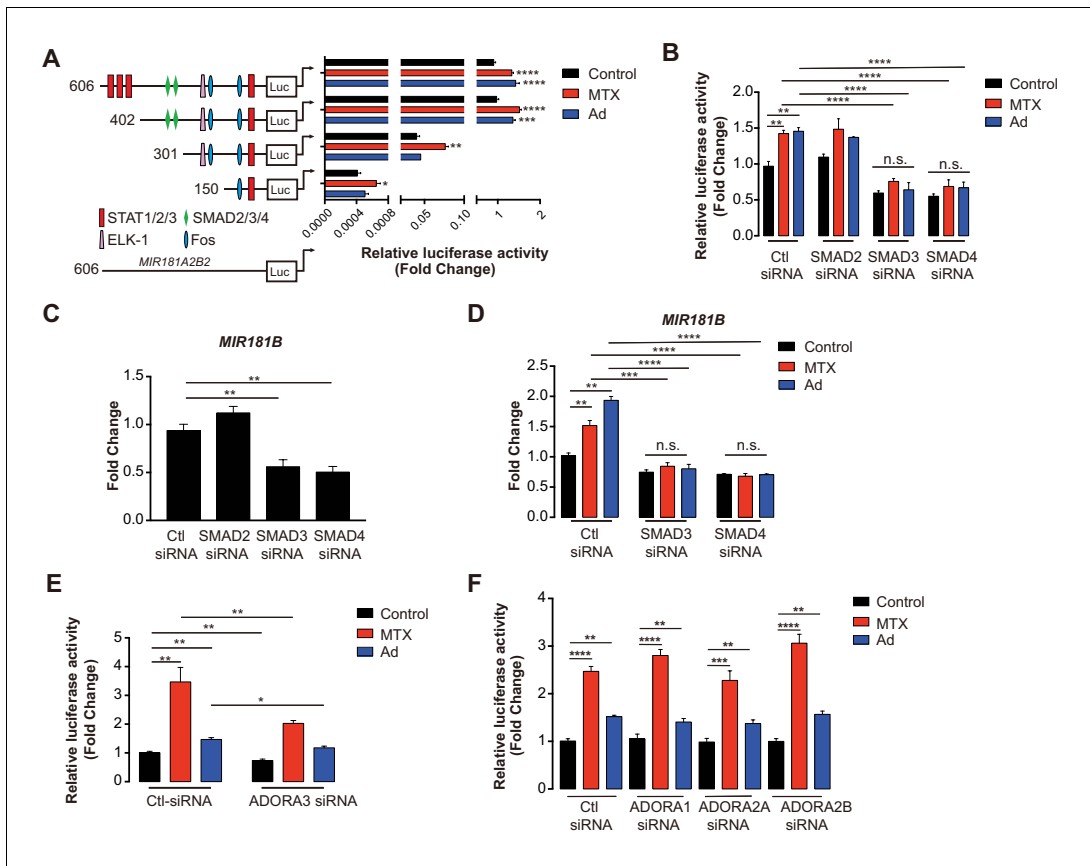

**Figure 3.** *MIR181A2B2* promoter analysis in response to methotrexate (MTX) or Ad in endothelial cells (ECs). (A) Luciferase reporter constructs containing the indicated (0.606–5.1 kb 5′upstream) *MIR181A2B2* promoter sequences were transfected in HEK 293 T cells and luciferase activity was measured 12 hr after treatment with MTX (10 µM) or Ad (50 µM), respectively. eight biological replicates. One-way ANOVA. (B) Luciferase reporter constructs containing the indicated (0.606 kb 5′upstream) *MIR181A2B2* promoter sequences were transfected in HEK 293 T cells and luciferase activity was measured 12 hr after treatment with MTX (10 µM) or Ad (50 µM), respectively. Seven to eight biological replicates. One-way ANOVA. (C) Effect of siRNA-mediated knockdown for *SMAD2, SMAD3, or SMAD4* in response to MTX or Ad on the 0.606 kb luciferase reporter. Three biological replicates. Unpaired two-tailed Student *t* test. (D–E) Real-time qPCR analysis of *MIR181B* expression in HUVECs transfected with siRNAs to negative control, *SMAD2, SMAD3, or SMAD4* in the (D) absence or (E) presence of MTX (10 µM) or Ad (50 µM) for 4 hr. three biological replicates. Unpaired two-tailed Student *t* test. (F–G) Luciferase reporters containing the 0.606 kb miR-181a2b2 promoter were transfected in combination with siRNA to negative control or *ADORA3* (F) or *ADORA1, ADORA2A, or ADORA2B* (G) and stimulated with MTX (10 µM) or Ad (50 µM). Please see *Figure 3—source data 1*. four biological replicates. Unpaired two-tailed Student *t* test. *p<0.05; **p<0.01; ***p<0.001; ****p<0.0001. All values represent mean ± SEM.

The online version of this article includes the following source data and figure supplement(s) for figure 3:

**Source data 1.**

**Figure supplement 1.** Silencing of adenosine receptor A1, A2A, or A2B did not affect methotrexate (MTX) or adenosine-induced *MIR181B* expression.

**Figure supplement 2.** The effect of methotrexate (MTX) and Ad on pri-*MIR181B-1* and pri-*MIR181B-2* transcript stability.

**Figure supplement 3.** The knockdown efficiency for *SMAD* siRNA.

## MTX improves insulin sensitivity and epididymal white adipose tissue (eWAT) inflammation in diet-induced obese mice, but not in Mir181a2b2[-/-] mice

Previous reports suggested that MTX can ameliorate diet-induced insulin resistance and inflammation (*DeOliveira et al., 2012*; *Myers et al., 2017*), although the mechanisms remained poorly

understood. To evaluate the effective dose of MTX on *Mir181b* expression in vivo, C57Bl6 mice were injected with 0.5 mg/kg or 1.0 mg/kg of MTX. Low-dose (i.e. 0.25 mg/kg) MTX showed no significant increase in *Mir181b* expression in plasma, liver, and intima, but 1 mg/kg did (*Figure 4—figure supplement 1A*). Moreover, C57Bl6 mice treated with adenosine exhibited significantly elevated *Mir181b* expression in plasma by 168% and was even more pronounced than high-dose MTX (*Figure 4—figure supplement 1B*). In accordance with this dose-response of MTX on *Mir181b* expression, we examined whether the effect of MTX (1 mg/kg) on insulin resistance and adipose tissue inflammation is dependent on *Mir181b2* expression in vivo. To this end *Mir181a2b2*$^{flox/flox}$ (flox) and *Mir181a2b2*$^{-/-}$ (KO) mice were placed on a 60% high-fat diet (HFD) for 12 weeks. Both flox and KO mice were injected with vehicle or MTX for 12 weeks (1 mg/kg/week by intraperitoneal injection), and insulin tolerance testing (ITT) and glucose tolerance testing (GTT) were performed (*Figure 4A*). As shown in *Figure 4B*, the body weights among the four groups were not significantly different, which suggest that the gain of body weight was independent of *Mir181a2b2* expression and MTX treatment. However, MTX treatment significantly improved insulin sensitivity and reduced the area under curves (AUC) for ITT by 23% (*Figure 4C*) compared with vehicle injected flox mice, but not in glucose tolerance (*Figure 4D*). In line with our previous report showing a protective effect of *Mir181b* mimics on insulin resistance (*Sun et al., 2016*), KO mice developed insulin resistance (IR) as shown by improved insulin sensitivity (increased AUC for ITT by 121%) and glucose tolerance (increased AUC GTT by 122%) (*Figure 4C,D*). Notably, the beneficial effect of MTX on insulin sensitivity was completely blocked in KO mice (*Figure 4C,D*). Moreover, flox mice treated with MTX had increased insulin signaling activity in eWAT and liver compared vehicle injected, indicated by an increase in phosphorylation of Akt (phospho-Akt) in eWAT (by 23%) and in liver (by 34%) (*Figure 4E*). In contrast, KO mice without MTX injections showed a significant reduction of Akt phosphorylation by 24% in eWAT and 44% in liver compared with control flox mice (*Figure 4E*). Moreover, the MTX-mediated upregulation of phospho-Akt in eWAT and liver was blocked in KO mice (*Figure 4E*). However, MTX did not increase phospho-Akt levels in skeletal muscle (SM) (*Figure 4E*). Taken together, these findings from systemic miR-181a2b2 KO mice suggest that MTX increases Akt phosphorylation in eWAT and liver through miR-181a2b2 expression, which contributes to improved insulin sensitivity.

To better characterize the underlying mechanisms how MTX and *Mir181a2b2* affects insulin resistance, we performed next-generation sequencing of RNA isolated from eWAT of these HFD-fed *Mir181a2b2* flox and KO mice with or without MTX treatment. Gene set enrichment analysis (GSEA) suggested that the most statistically significant [false discovery rate (FDR) < 0.5] altered genes in eWAT from *Mir181a2b2* KO mice were linked to cell adhesion (e.g. platelet-endothelium-leukocyte interactions, cell-matrix interactions, and cell junctions), chemotaxis, and immune response (*Figure 4—figure supplement 2A-B*). In eWAT from MTX-treated *Mir181a2b2* flox and KO mice, GSEA found that cell cycle, inflammation, and chemotaxis pathways were more affected by MTX (*Figure 4—figure supplement 2C-D*). Macrophage accumulation plays a critical role in obesity-associated IR (*Chawla et al., 2011*; *McNelis and Olefsky, 2014*) and overexpression of *Mir181b* did not directly regulate cell-intrinsic monocyte/macrophage migration, proliferation, and activation (*Sun et al., 2016*). Moreover, we confirmed that MTX administration reduced *VCAM-1* mRNA expression by 36% in eWAT in flox mice, but not MTX-injected KO mice (*Figure 4F,G*). In line with reduced *VCAM-1* expression, we observed that macrophage content, as indicated by Mac2 staining, was significantly reduced by 80% in eWAT in MTX-injected flox mice (*Figure 4H*). In contrast, KO mice treated with MTX showed no inhibitory effect on *VCAM-1* mRNA expression and macrophage content in eWAT (*Figure 4F,G*). Taken together, these results suggest that MTX administration leads to reduced EC activation and macrophage accumulation in white adipose tissue, largely dependent on *Mir181a2b2* expression.

## MTX improves insulin sensitivity in obese mice, but not in EC-specific Mir181a2b2 KO mice

It is well-established that some miRNAs may have a cell-type and/or tissue-specific expression and function which can impact the regulation of gene networks (*Gao et al., 2011*; *Sood et al., 2006*; *Thomou et al., 2017*; *Zhou et al., 2013*). *Mir181b* plays a dominant role in the vascular endothelium (*Lin et al., 2016*; *Sun et al., 2014a*; *MICU Registry et al., 2012*; *Sun et al., 2016*); however, its EC-specific genetic deletion has not been previously described. To assess whether endothelial

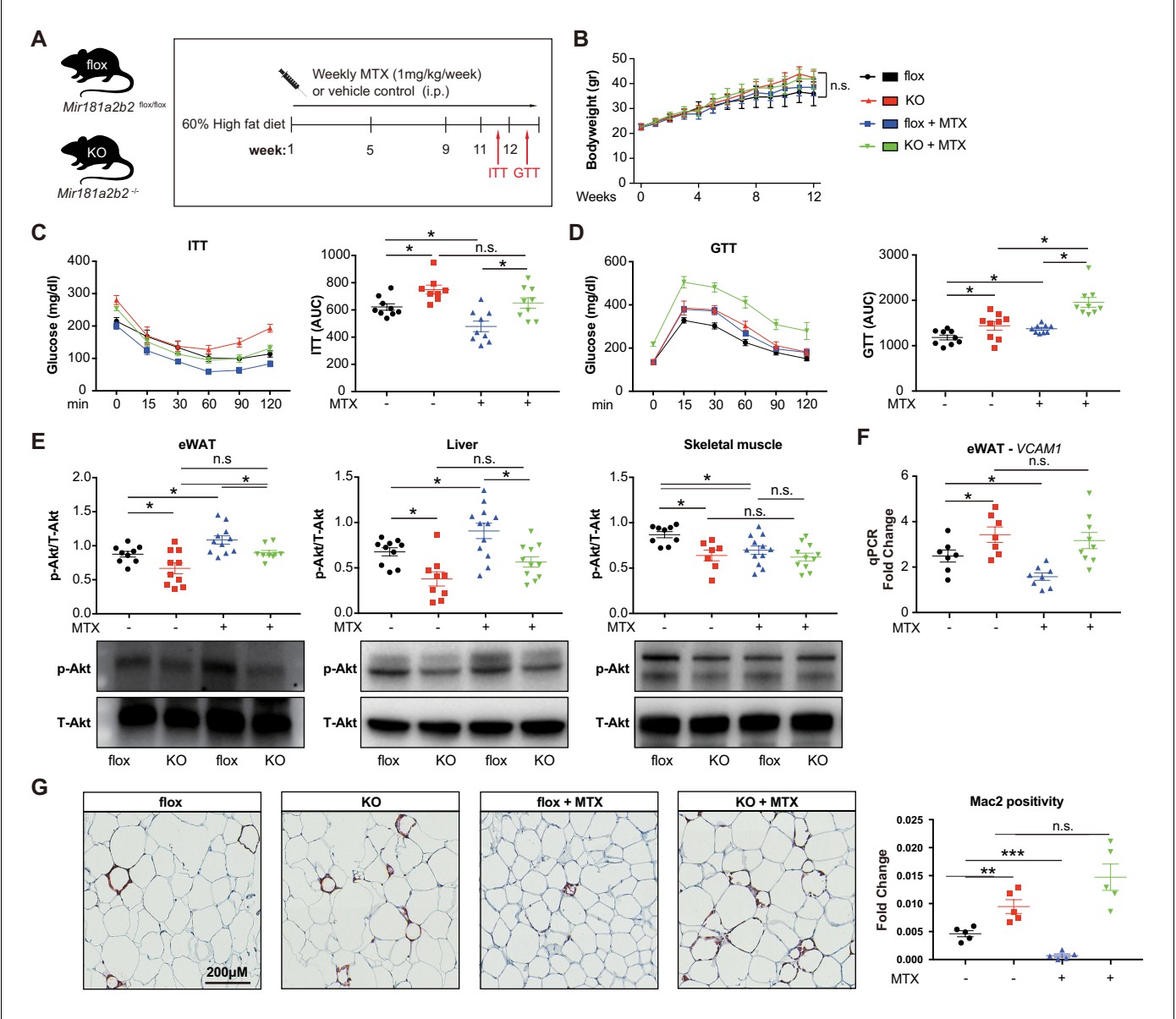

**Figure 4.** Systemic *Mir181a2b2* KO blocks methotrexate (MTX)-mediated insulin sensitivity and visceral fat inflammation in diet-induced obesity. (**A**) Schema of experimental procedure for *Mir181a2b2* ^flox/flox^ (flox) and *Mir181a2b2*^−/−^ (KO) mice that were placed on a 60% high-fat diet (HFD) for 12 weeks. Each group of mice was weekly i.p. injected with vehicle control or MTX (1 mg/kg). (**B**) Body weights were measured weekly. Blood glucose levels were measured at week 11 for (**C**) insulin tolerance testing (ITT) and on week 12 for (**D**) glucose tolerance testing (GTT) with calculated area under the curves (AUC), respectively. (**E**) Western blot analysis of Akt and pSer473-Akt in epididymal white adipose tissue (eWAT), liver, and skeletal muscle with quantification across n = 3 independent experiments. (**F**) Real-time qPCR analysis of VCAM-1 expression in eWAT. (**B–F**), n = 9–10 mice per group, one-way ANOVA. (**G**) Paraffin sections of eWAT were stained with Mac2 and the positive areas were quantified, n = 5 mice per group, one-way ANOVA. *p<0.05; **p<0.01; ***p<0.001. All values represent mean ± SEM.

The online version of this article includes the following figure supplement(s) for figure 4:

**Figure supplement 1.** MTX and Ad increase *Mir181b* expression in the circulation and tissues.

**Figure supplement 2.** Gene set enrichment analyses of eWAT from *MiR181a2b2* flex and KO mice.

*Mir181a2b2* is mediating the observed phenotypes with respect to MTX and insulin sensitivity, we generated inducible EC-specific *Mir181a2b2* mice (iEC-KO) by breeding the *Mir181a2b2*^flox/flox^ mice to tamoxifen-inducible endothelial-specific Cre (vascular endothelial cadherin promoter [VECad-Cre-ER^T2^]) (***Figure 5A***). Compared to vehicle control injected mice, *Mir181b* expression in tamoxifen

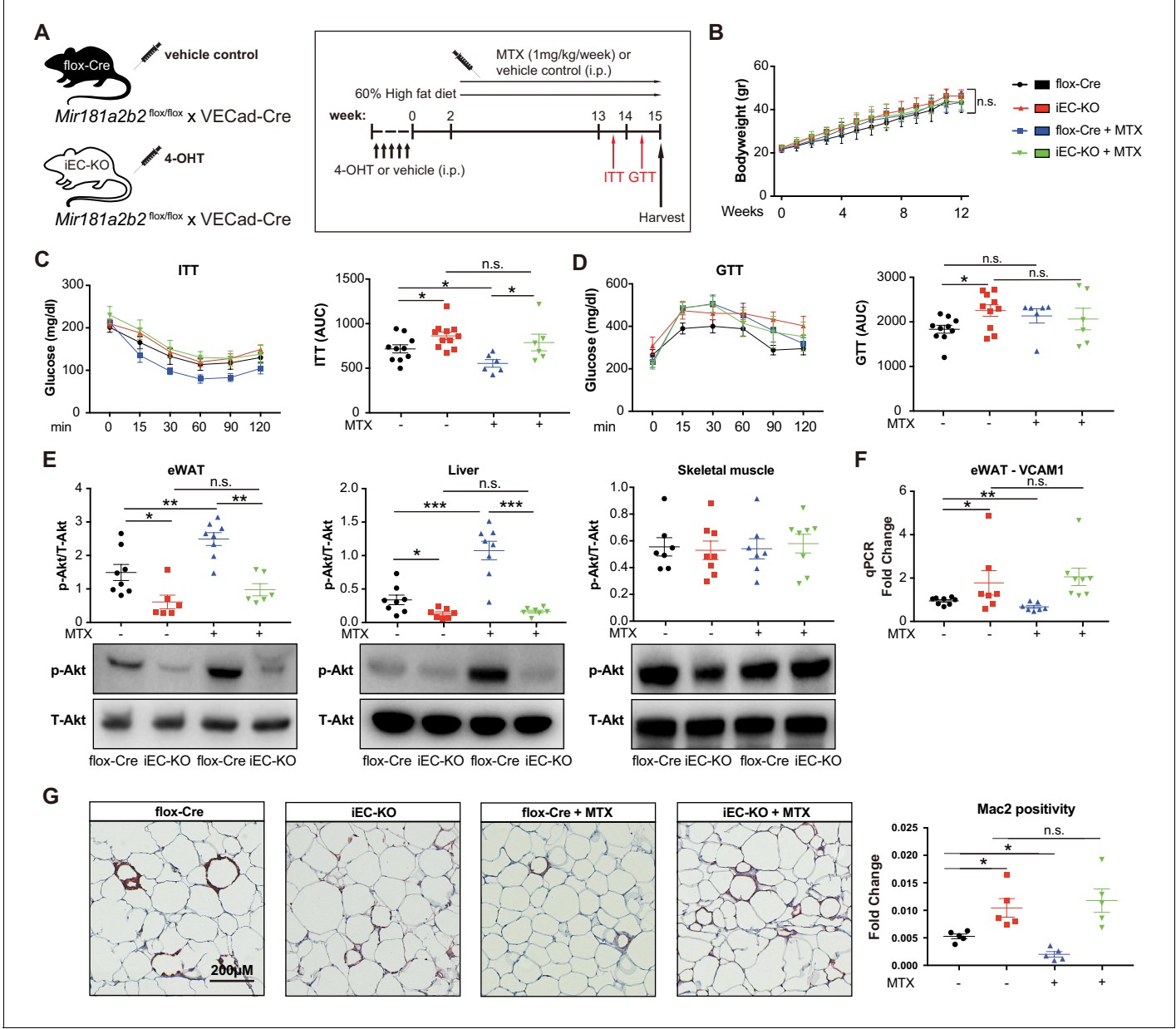

**Figure 5.** Endothelial cell (EC)-specific *Mir181a2b2* KO blocks methotrexate (MTX)-mediated insulin sensitivity and visceral fat inflammation in diet-induced obesity. (**A**) Schema of experimental procedure for control *Mir181a2b2*flox/flox; VECad-Cre (flox-Cre) or tamoxifen injected EC-specific *Mir181a2b2* KO (iEC-KO) mice that were placed on a 60% high-fat diet (HFD) for 15 weeks. Each group of mice was weekly i.p. injected with vehicle control or MTX (1 mg/kg). (**B**) Body weights over time of mice treated with vehicle or MTX, respectively. (**C–D**) ITT (**C**) and GTT (**D**) were measured and AUCs were quantified for each group. (**E**) Western blot analysis of Akt and pSer473-Akt in eWAT, liver, and skeletal muscle tissues. (**F**) Real-time qPCR analysis of VCAM-1 expression in eWAT. (**G**) Paraffin sections of eWAT were stained with Mac2 and the positive areas were quantified, n = 5 mice per group, one-way ANOVA. (**B–G**), n = 6 to 10 mice per group, one-way ANOVA. *p<0.05; **p<0.01; ***p<0.001. All values represent mean ± SEM. The online version of this article includes the following figure supplement(s) for figure 5:

**Figure supplement 1.** Transcriptomic changes in eWAT of systemic *Mir181a2b2* KO mice in response to high-fat diet (HFD) and methotrexate (MTX).
**Figure supplement 2.** Breeding schema for creating endothelial-specific *Mir181a2b2* KO mouse strain.
**Figure supplement 3.** Western blot analysis of SMADs expression in eWAT from high-fat diet (HFD)-fed flox-Cre and iEC-KO mice treated with or without methotrexate (MTX) for 12 weeks.

injected iEC-KO mice was reduced by 60% and 80% in aortic intima and ECs isolated from SM, respectively, but not in the aortic media and non-EC fraction of SM (*Figure 5—figure supplement 1B–E*). To assess whether endothelial *Mir181a2b2* is driving the diet-induced IR and mediates MTX's protective effects, we first examined MTX regulation of *Mir181b* expression. While *Mir181b* was reduced in the aortic intima of iEC-KO mice compared to flox-Cre, MTX increased *Mir181b* in flox-Cre, but not in iEC-KO mice (*Figure 5—figure supplement 2A*). Consistent with upregulated *Mir181b* expression, MTX reduced *VCAM-1* expression in the intima of flox-Cre mice, but not in iEC-KO mice (*Figure 5—figure supplement 2B*). Because we identified that SMAD3/4 signaling contributed largely to *Mir181b2* expression (*Figure 3*), we next investigated expression of SMADs in eWAT tissues from these HFD-fed flox-Cre and iEC-KO mice. We found that *SMAD2* and *SMAD3* expression decreased in MTX-treated iEC-KO mice compared to flox-Cre mice, whereas there were no differences between flox-Cre and iEC-KO mice without MTX treatment, suggesting a feedback mechanism involved in the MTX-SMAD-Mir181b cascade (*Figure 5—figure supplement 3*).

As shown in *Figure 5B*, there were no significant differences in body weights between iEC-KO and control mice treated with or without MTX after 12 weeks of HFD. However, MTX administration significantly improved insulin sensitivity by 23% in ITT of control flox-Cre mice (*Figure 5C*), whereas MTX had no effect on insulin sensitivity in iEC-KO mice (*Figure 5C*). Similar to *Mir181a2b2* systemic KO and control mice, MTX administration did not improve glucose tolerance either in iEC-KO or control mice (*Figure 5D*). Moreover, the MTX-mediated induction of Akt phosphorylation in eWATs and livers was blocked in iEC-KO mice (*Figure 5E*). Furthermore, MTX reduced *VCAM-1* expression in eWAT in flox-Cre control mice, but not in iEC-KO mice (*Figure 5F*). Consistent with previous results (*Figure 4H*), MTX administration significantly reduced Mac2 positive macrophage content in eWAT tissues in flox-Cre mice, whereas no effect was observed in iEC-KO mice (*Figure 5G*). Finally, iEC-KO mice without MTX treatment showed a significant increase in Mac2 positive macrophages compared to flox-Cre control mice in eWAT tissues (*Figure 5G*). Taken together, these results identify a crucial role of endothelial *Mir181a2b2* for mediating MTX's anti-inflammatory effects.

## Discussion

Low-grade inflammation in adipose tissue plays a critical role in the pathogenesis of obesity-associated insulin resistance, an important risk factor for the development of cardiovascular disease and type 2 diabetes (*Arner and Kulyté, 2015*). The Canakinumab Atherosclerosis Outcome Thrombosis Study (CANTOS) demonstrated that targeting inflammation reduces cardiovascular events by targeting interleukin-1β, independent of lipid lowering (*Ridker et al., 2017*). In contrast, MTX, an anti-inflammatory drug that is widely used to treat patients with RA and psoriatic arthritis, did not reduce inflammatory markers, such as interleukin-1, interleukin-6, C-reactive protein, or cardiovascular events in the CIRT trail (*Ridker et al., 2019*). The molecular mechanisms of how MTX reduces inflammation remains poorly understood. Previous studies found that Ad was the major downstream anti-inflammatory effector of MTX. By decreasing the catabolism of adenosine, MTX increases extracellular Ad, which in turn mediates its cellular effects via engagement of specific receptors ($A_1$, $A_{2A}$, $A_{2B}$, and $A_3$; *Cronstein and Sitkovsky, 2017*). Among these adenosine receptors, $A_{2A}$ and $A_3$ are the most important receptors for mediating adenosine's cellular anti-inflammatory effect (*Cronstein et al., 1993*; *Montesinos et al., 2003*).

MiRNAs are key post-transcriptional gene regulators and even moderate changes in miRNA expression can have profound functional impact in the vascular endothelium in response to divergent acute and chronic inflammatory conditions. For instance, we and others found that *Mir181b* expression was reduced by 40–60% in the vascular endothelium under atherosclerosis, sepsis, thrombosis, insulin resistance, and other conditions (*Guo et al., 2018*; *Lin et al., 2016*; *Ma et al., 2016*; *Sun et al., 2014a*; *MICU Registry et al., 2012*; *Sun et al., 2016*). Overexpression of *Mir181b* by tail vein injection of *Mir181b* mimics (~1.5- to 2.5-fold increase in Mir181b expression) improved acute vascular inflammation induced by endotoxemia (*MICU Registry et al., 2012*), and chronic vascular inflammation triggered by atherosclerotic plaque formation (*Sun et al., 2014a*), obesity and insulin resistance (*Sun et al., 2016*), and thrombin-induced arterial thrombosis (*Lin et al., 2016*). In ECs, *Mir181b* inhibits NF-κB signaling by targeting importin-α3 or caspase recruitment domain family member 10 (Card10), whereas it activates phospho-Akt by targeting, pleckstrin homology domain leucine-rich repeat protein phosphatase isozyme2 (PHLPP2) (*Lin et al., 2016*; *Sun et al., 2014a*;

*MICU Registry et al., 2012*; *Sun et al., 2016*). Moreover, using an E-selectin-targeting microparticles system to deliver *Mir181b* specifically to the activated vascular endothelium, *Mir181b* significantly reduced atherosclerosis (*Ma et al., 2016*). However, little is known how this anti-inflammatory Mir181b is regulated in ECs. While acute or chronic pro-inflammatory stimuli reduced *Mir181b* expression (*Sun et al., 2014a*; *MICU Registry et al., 2012*; *Sun et al., 2016*), factors or signaling pathways that increase its expression in the vascular endothelium have not been defined.

Ad is continuously released by cells to the extracellular environment at a low concentration (less than 1 µM) in unstressed tissues, whereas Ad concentrations can reach as high as 100 µM in inflamed or ischemic tissues (*Haskó and Cronstein, 2004*; *Jacobson and Gao, 2006*). In addition, ECs are a major source of extracellular Ad at sites of metabolic distress, inflammation, and infection (*Cronstein, 1994*). In agreement with a recently published report (*Xu et al., 2017*), we found that Ad potently repressed TNF-α-induced inflammatory genes expression in an *ADORA3*-dependent manner, but not the other three Ad receptors. Moreover, we demonstrated that the MTX and Ad-mediated protective effects on ECs were dependent on Mir181b expression. Interestingly, silencing *ADORA3* decreased Mir181b expression, and blocked the TNF-α-mediated repression of Mir181b expression and abrogated the MTX- and Ad-mediated induction of *Mir181b* expression. Indeed, after TNF-α stimulation, the expression of adenosine kinase (ADK), a major intracellular adenosine metabolic enzyme, is increased and thereby results in decreased intracellular adenosine concentration, which in turn reduces extracellular adenosine (*Xu et al., 2017*). Taken together, these data suggest that the MTX- and Ad-mediated induction of *Mir181b* expression is largely dependent on *ADORA3* expression. Moreover, the TNF-α-repression of *Mir181b* expression may be through a similar integrated network by inhibiting adenosine production and release and, in turn, *ADORA3* signaling.

Systemic *MiR181a2b2* KO and endothelial-specific-deficient *Mir181a2b2* mice exhibited EC activation, vascular inflammation, and insulin resistance in obese mice, suggesting that endothelial *Mir181a2b2* plays a key role in EC activation and progression of insulin resistance. Moreover, the anti-inflammatory effects and improvement in insulin sensitivity by MTX were dependent on endothelial *Mir181a2b2*. Mechanistically, MTX activated an Ad-*ADORA3*-SMAD3/4 signaling cascade that transcriptionally activated *MIR181A2B2* expression and modestly increased pri-*MIR181B2* mRNA stability at the post-transcriptional level. In light of these findings, the current study indicates that endothelial *Mir181a2b2* plays a critical role in diet-induced insulin resistance and mediates in part MTX's anti-inflammatory effects. Moreover, our data suggest that high-dose of MTX (1 mg/kg) is essential to increase *Mir181b* expression and modulate a large gene network involving EC-leukocyte adhesion, chemotaxis, immune response, and cell cycle (*Figure 4—figure supplement 1*) in eWAT; consequently, this may ameliorate EC activation, insulin resistance, and adipose tissue inflammation. Several of these identified pathways (EC-leukocyte adhesion, chemotaxis, and immune response) are consistent with pathways known to be regulated by Mir181b (*Guo et al., 2018*; *Lin et al., 2016*; *Ma et al., 2016*; *Sun et al., 2014a*; *MICU Registry et al., 2012*; *Sun et al., 2016*). In addition, *Mir181a2b2* KO mice exhibited more glucose intolerance under basal conditions compared to flox mice, and this response was further exacerbated by MTX treatment (*Figure 4D*). While the mechanisms underlying this exaggerated response in the KO mice is not clear, we cannot rule out a participatory role of a broad range of tissues including the pancreas (*Baghdadi, 2020*). In our prior study of HFD-induced obese mice, systemic delivery of *Mir181b* mimics (to overexpress *Mir181b*) improved insulin resistance and increased phospho-AKT in eWAT but not in liver or skeletal muscle (*Sun et al., 2016*). In this current study, MTX treatment potently increased phospho-Akt in liver, eWAT, and skeletal muscle in control mice (*Figure 4E*), but not in those tissues from *Mir181a2b2*$^{-/-}$ mice or in liver or eWAT in iECKO mice (*Figure 5E*), suggesting that deficiency of miR-181a2b2 may figure more prominently in these tissues and underlie differences for phospho-AKT expression.

Previous studies demonstrated that MTX at a dose of 1–4 mg/kg more potently reduced inflammation (*DeOliveira et al., 2012*; *Thornton et al., 2016*). For example, the MUSICA trial demonstrated that high-dose MTX reduced inflammation (including circulating hsCRP) and RA disease activity better than low-dose MTX (*Kaeley et al., 2018*). Our findings may also inform the lack of efficacy and anti-inflammatory effects of low-dose MTX in the CIRT study. Given the side effect profile of MTX including higher rates of mouth sores or stomatitis, liver enzyme elevations, mild leukopenia, and non-basal-cell skin cancers (*Ridker et al., 2019*), future studies that therapeutically target signaling downstream to MTX (e.g. the adenosine-ADORA3-SMAD3/4-*MIR181B* signaling pathway)

may leverage its anti-inflammatory effects potentially without its dose-limiting toxicity or side effects.

There are several limitations to our study. We cannot definitively bifurcate the contribution of *Mir181b1* or *Mir181b2* to EC activation and vascular inflammation. Mir181b is expressed from two distinct loci, that is the *Mir181a-1/Mir181b-1* cluster and the *Mir181a-2/Mir181b-2* cluster (*Sun et al., 2014b*). In addition, *Mir181a-1, Mir181a-2, Mir181b-1, and Mir181b-2*, each may have its unique functions, making a complicated modulation of large gene networks. Although we identified that the primary transcript of *MIR181B-2* is more likely responsive to TNF-α stimulation in HUVECs and isolated primary mouse lung ECs using the systemic and endothelial-specific deficient *Mir181a2b2* mice, a recent study found that systemic double knockout of *Mir181a1b1* and *Mir181a2b2* improved HFD-induced obesity and insulin resistance (*Virtue et al., 2019*), raising the possibility that targeting *Mir181a1b1* may impact distinct pathways and functions than *Mir181a2b2*. Future studies will be required to further characterize more details of these individual loci and responsiveness to diverse stimuli. In addition, while we found that MTX rescues *Mir181b* expression in HFD-fed mice, future studies will be of interest to validate MTX effects under chow-fed diet conditions. We also found that *SMAD2* and *SMAD3* expression decreased in eWAT of MTX-treated iEC-KO mice compared to flox-Cre mice only in response to MTX treatment. Future studies will need to clarify the mechanisms underlying a possible positive feedback loop involving the MTX-SMAD-*MIR181B* signaling pathway. Finally, while tamoxifen has been reported to affect GTT at early time points of 1–3 weeks after injection (*Ceasrine et al., 2019*), in the current study we performed GTT and ITT at 12 weeks after initial injection of the tamoxifen metabolite 4-hydroxytamoxifen, making the likelihood of impact from 4-OHT extremely low as reported (*Liu et al., 2015*; *Payne et al., 2018*; *Tian et al., 2013*; *Zhang et al., 2016*).

In summary, we have identified that endothelial *Mir181a2b2* serves as a key regulator of pro-inflammatory stimuli-mediated EC activation, HFD-induced insulin resistance, and WAT inflammation. We also found that the anti-inflammatory effect of MTX was dependent on miR-181a2b2 expression in ECs, and that there is a positive correlation between the dose of MTX and *Mir181b* expression in circulating plasma and ECs. MTX prevented cytokine-induced EC activation through a unique downstream adenosine-*ADORA3*-SMAD-*MIR181B* signaling cascade. Therefore, strategies targeting this pathway to increase the expression of *Mir181b* in ECs may provide a novel translational approach to limit vascular inflammation in a range of chronic disease states.

## Materials and methods

### Cell culture and transfection

HUVECs were purchased from Lonza (cc-2159) and cultured in EC growth medium EGM-2 (cc-3162). HEK 239 T cells (CRL-3216, ATCC) were cultured in DMEM supplemented with 10% FBS and 1% penicillin and streptomycin. Cells were passaged less than five times for all experiments. Cell types were authenticated as follows: HUVECs, morphology; HEK293T, cell size and efficiency in transfection reporter assays. All cell lines were tested mycoplasma-free. Primary lung ECs from mice were isolated and cultured in as described (*Sun et al., 2016*). Briefly, isolated lungs from miR-181a2b2$^{flox/flox}$ and miR-181a2b2$^{-/-}$ mice were grinded with scissors ($1 \times 2$ mm$^2$) and digested with collagenase type II (Worthington) and dispase (Roche) (1 mg/ml each in DMEM/F12). Digestion was neutralized with DMEM/F12 medium containing 10% FBS, followed by centrifugation at 500 g for 10 min at 4°C and re-suspend in incubation buffer (PBS pH 7.2, 0.1% BSA, 2 mM EDTA, 0.5% FBS). ECs were captured using 10 µl anti-PECAM-1 (1:12.5, 557355, BD Pharmingen) antibody-conjugated sheep anti-rat IgG Dynabeads (00412289, Invitrogen) mixed with buffer containing the re-suspended cell pellet and incubated for 20 min at 4°C. After incubation, Dynabeads were mounted using the Magnetic Separation Rack and washed five times using wash buffer (PBS pH 7.2, 0.1% BSA) and cultured in gelatin (0.1%) pre-coated plates in EC medium (M1168, Cell Biologics) with kit (M1168-kit, Cell Biologics). EC identity was verified by both morphology and expression for CD31 by flow cytometry. HUVECs or primary lung ECs were plated on 12-well plates at 60,000/well or 150,000/well and allowed to grow to 80–90% confluency under growing conditions or 70–80% for transfection. Cells were pretreated with MTX (0.1, 1, 10 or 100 µM, M8407, Sigma), adenosine (1, 5, 10, 50 or 100 µM, A4036, Sigma), atorvastatin (1 µM, 1044516, Sigma), mevastatin (1 µM, M2537, Sigma), simvastatin

(1 μM, S6169, Sigma), or rosuvastatin (1 μM, SML1264, Sigma) for 3 hr and stimulated with or without 10 ng/ml of recombinant human TNF-α (210-TA/CF, R and D Systems) or 20 ng/ml recombinant mouse TNF-α (410-MT/CF, R and D Systems) for various times, according to the experiment: Western blot, 8 hr; real-time qPCR, 1, 4, 7, or 12 hr. Lipofectamine 2000 transfection reagent (11668019, Invitrogen) was used for transfection, following the manufacturer's instructions. On-Target plus human control siRNA-Smart poll, On-Target plus human AdoRA1 siRNA-Smart poll (L-005415–00, Dharmacon), On-Target plus human AdoRA2A siRNA-Smart poll (L-005416–00, Dharmacon), On-Target plus human AdoRA2B siRNA-Smart poll (L-005417–00, Dharmacon) or On-Target plus human *ADORA3* siRNA-Smart poll (L-005418–00, Dharmacon) were transfected at 100 nM. SiRNA control (AM4636) and validated siRNA including *SMAD2* (VHS41106), *SMAD3* (VHS41111), *SMAD4* (s8403) from Invitrogen were transfected at 20 nM. For co-transfection studies, On-Target plus human control siRNA-smart poll (100 nM), On-Target plus human *ADORA3* siRNA-smart poll (100 nM), miRNA negative control (10 nM, AM17110, Ambion), or pre-Mir181b (10 nM, PM12442, Ambion) were transfected as indicated in respective experiments. HUVECs were treated with 10 ng/ml recombinant human TNF-α 36 hr post-transfection at indicated times points.

## Luciferase activity assay and cell culture transfection

HEK 293 T cells were plated on 12-well plates at 300,000/well up to 70–80% confluency and transfected with 500 ng of the indicated *Mir181a2b2* promoter reporter constructs (Supplementary file 1) and 10 ng Renilla plasmid (E2231, Promega). Adenosine receptor siRNAs and SMAD2/3/4 siRNAs were co-transfected at 100 or 20 nM final concentration and cells were treated with MTX (10 μM) or Adenosine (50 μM) 24 hr post-transfection for 12 hr. Transfected cells were collected in 200 μl reporter lysis buffer (E1910, Promega) and luciferase activity was measured using a Dual-Luciferase reporter assay system (E1910, Promega). Each reading of luciferase activity was normalized to the Renilla activity.

## mRNA stability assay

For mRNA stability assays, HUVECs were incubated with 10 μM MTX or 50 μM Ad with 10 μg/ml Actinomycin D (SBR00013, Sigma) to inhibit transcription. At the indicated time points after the addition of Actinomycin D, cells were harvested and total RNA was extracted. The expression levels of *pri-Mir181b1* and *pri-Mir181b2* at each time point were measured by real-time qPCR as described below and normalized to the according Hprt levels. The remaining mRNA was determined by comparison with the expression level of the relevant gene at the zero time point (designated 100%) when Actinomycin D was added.

## Animal studies

*Mir181a2b2*<sup>flox/flox</sup> and *Mir181a2b2*<sup>−/−</sup> mice were used as previously described (*Henao-Mejia et al., 2013*) Mice expressing tamoxifen-inducible endothelial-specific VE-cadherin (VECad-Cre-ER<sup>T2</sup>; also known as Cdh5(PAC)-CreER<sup>T2</sup>) was kindly provided by R. Adams (*Wang et al., 2010*). C57BL/6 mice were purchased from Charles River Laboratories (#027). Inducible EC-specific *Mir181a2b2*-deficient mice (*Mir181a2b2* <sup>flox/flox</sup>; VECad-Cre-ER<sup>T2</sup>) was generated by crossbreeding *Mir181a2b2*<sup>flox/flox</sup> mice and VECad-Cre-ER<sup>T2</sup> mice. For induction of Cre activity, four weeks old male *Mir181a2b2*<sup>flox/flox</sup> mice carrying the VECad-Cre-ER<sup>T2</sup> transgene were treated with either 4-hydroxytamoxifen (H6278, Sigma) (10 mg/kg, i.p.) or the same volume of vehicle for 5 consecutive days to generate EC-specific *Mir181a2b2*-deficient mice (*Mir181a2b2*<sup>iECKO</sup>) and flox-Cre control mice (*Mir181a2b2*<sup>ECfl/fl</sup>). For measurement of *Mir181b* in plasma, 8 weeks male C57BL/6 mice were tail vein injected with MTX (0.25 mg/kg or 1 mg/kg) or adenosine (4 mg/kg) for 3 hr and blood was collected in EDTA-containing tubes. Plasma samples were generated by centrifugation at 3000 rpm for 10 min and stored at −80° C. Total RNA was isolated from plasma by using the Total RNA Purification Kit from Norgen Biotek Corporation as described in our previous studies (*Sun et al., 2014a*; *MICU Registry et al., 2012*; *Sun et al., 2016*). Male mice were age-matched in all experiments and cage-matched littermates were used for experiments. Analyses of in vivo samples were performed by blinded observers. All mice were maintained under SPF conditions at an American Association for the Accreditation of Laboratory Animal Care-accredited animal facility at the Brigham and Women's Hospital. All animal protocols were approved by the Institutional Animal Care and Use Committee at Harvard Medical

School, Boston, MA and conducted in accordance with the National Institutes of Health Guide for Care and Use of Laboratory Animals.

## Diet-induced obesity, MTX administration, and glucose and insulin tolerance testing

*Mir181a2b2*<sup>flox/flox</sup>, *Mir181a2b2*<sup>−/−</sup>, *Mir181a2b2*<sup>ECfl/fl</sup> and *Mir181a2b2*<sup>iECKO</sup> mice were injected i.p. with either MTX (M8407, Sigma, 1 mg/kg/week) or the same volume of vehicle control and placed on a high-fat diet (HFD) for 12–15 weeks containing 60 kcal% fat (D12492, Research Diets). Insulin tolerance testing (ITT) and glucose tolerance testing (GTT) were performed after HFD treatment as described in our previous study (*Sun et al., 2016*). Briefly, for GTT, mice were fasted for 12 hr and then injected i.p. with D-glucose (G7201, Sigma, 1 g/kg). ITT was performed on mice after 6 hr fasting and injected i.p. with recombinant human regular insulin (0.75 U/kg). Blood glucose levels were measured before injection and at 15, 30, 60, 90, and 120 min after glucose or insulin injection using One Touch Ultra glucometer (LifeScan).

## Immunohistochemistry (IHC)

IHC was performed as described in our previous study (*Sun et al., 2016*). Briefly, tissues were fixed with neutral buffered 10% formalin solution (HT501128, Sigma), embedded in paraffin wax, cut into sections at 6 µm, and deparaffinized. Antigen retrieval was performed using Bond ER1 (AR9961, Leica) for 30 min. Sections were incubated with anti-Mac-2 (553322, BD Pharmingen) in 1:900 dilutions for 90 min at room temperature. Primary antibodies binding to tissue sections was visualized using Bond Polymer Refine Detection kit (DS9800, Leica), and counterstained with hematoxylin. Representative images were captured by a digital system, and the Mac-2-postive areas in each section were determined by detecting the staining intensity with computer-assisted image analysis software (Image-Pro Plus, Media Cybernetics) and data were presented as a ratio of positive area to tissue area. Data were analyzed in a blinded fashion by two independent observers.

## Intima RNA isolation from aorta tissue

Isolation of intima RNA from aorta was performed as described in our previous studies (*Sun et al., 2014a*; *Sun et al., 2016*). In brief, aortas were carefully flushed with PBS, followed by intima peeling using TRIzol reagent (15596018, Invitrogen). TRIzol was flushed for 10 s −10 s pause - another 10 s flushed and collected in an Eppendorf tube (~300–400 µL total) and snap frozen in liquid nitrogen.

## RNA isolation and real-time qPCR

Tissues were homogenized using TissueLyser II (Qiagen) according to manufacturer's instructions. For RNA isolation TRIzol reagent (Invitrogen) or RNeasy kit (Qiagen) was used based on manufacturers protocol. Subsequent RT-qPCR was performed using High-Capacity cDNA Reverse Transcription kit (4368813, Applied Biosystems). For SyberGreen-based assay GoTaq qPCR Master Mix (A6001, Promega) was used; and for TaqMan Universal Master Mix II, UNG (4440038, Life Technologies) was used. Expression of mRNAs and miRNA expression were normalized to Hprt, β-actin, or U6 (Aglient, AriaMx Real-Time PCR System). Specific primers including Mir181b-5p (#001098), U6 (#001973), human-primary-miR-181a1b1 (#Hs03302966_pri), human-primary-miR-181a2 (#Hs03302889_pri), human-primary-Mir181b1 (#Hs03302963_pri), for TaqMan system were purchased from Life Technologies. Changes in expression were calculated using deltadelta Ct method. Primer sequences are described in (Supplementary file 2).

## RNA-Seq analysis

TRIzol reagent was used for RNA isolation from eWAT tissues based on manufacturers' protocol. RNA-Seq analysis was performed after ribodepletion and standard library construction using Illumina HiSeq2500 V4 $2 \times 100$ PE (Genewiz, South Plainfield, NJ). All samples were processed using an RNA-seq pipeline implemented in the bcbio-nextgen project (https://bcbio-nextgen.readthedocs.org/en/latest/). Raw reads were examined for quality issues using FastQC (http://www.bioinformatics.babraham.ac.uk/projects/fastqc/) to ensure library generation and sequencing were suitable for further analysis. Trimmed reads were aligned to UCSC build mm10 of the Mouse genome, augmented with transcript information from Ensembl release 79 using STAR. Alignments were checked

for evenness of coverage, rRNA content, genomic context of alignments (for example, alignments in known transcripts and introns), complexity, and other quality checks using a combination of FastQC, Qualimap. Counts of reads aligning to known genes were generated by featureCounts (*Liao et al., 2014*). Differential expression at the gene level were called with DESeq2. The total gene hit counts and CPM values were calculated for each gene and for downstream differential expression analysis between specified groups was performed using DESeq2 and an adapted DESeq2 algorithm, which excludes overlapping reads. Genes with adjusted FDR < 0.1 and log2 fold-change (1.5) were called as differentially expressed genes for each comparison. Mean quality score of all samples was 36.12 within a range of 22,000,000–31,000,000 reads per sample. All samples had at least >70% of mapped fragments over total. MetaCore (v20.2) was used for gene set enrichment analysis. RNA-Seq data are available through the Gene Expression Omnibus (GSE164251).

## Western blot

Tissues were homogenized using TissueLyser II (Qiagen) according to manufacturer' instructions. Proteins were isolated using RIPA buffer (Boston BioProdcuts, BP-115) with protease inhibitor and phosphatase inhibitors. Protein concentrations were determined using Pierce BCA assay (Thermo Scientific). Twenty µg protein were loaded per lane on a 4–20% Mini-PROTEAN TGX Gel (Bio-Rad, 456–1096). Separated proteins were transferred to PVDF membranes using the Transfer Turbo Blot system (Bio-Rad) and Trans-Blot Turbo RTA Transfer Kit (Bio-Rad, 170–4272). The membrane was blocked with 5% nonfat milk in TBST for 1 hr at room temperature. After blocking, the membrane was incubated overnight at 4°C with antibodies against *ADORA1* (1:500, ab82477, abcam), *ADORA2A* (1:750, ab3461, abcam), *ADORA2B* (1:750, ab40002, abcam), *ADORA3* (1:500, ab203298, abcam), *VCAM-1* (1:1000, ab134047, abcam), *ICAM-1* (1:3000, BBA3, R and D Systems), *E-Selectin* (1:1000, BBA16, R and D Systems), p-AKT (1:1000, 4060, Cell Signaling), T-AKT (1:1000, 2920, Cell Signaling), β-actin (1:4000, 4970, Cell Signaling). Quantification of protein bands were performed using a luminescent image analyzer (BioRad, Chemidoc).

## Statistics

Statistical analyses were performed using GraphPad Prism version 7.0 (GraphPad Software, Inc, CA). Unpaired two-tailed Student *t* test was used to determine statistical significance between two groups. Multiple groups were analyzed by using One-way ANOVA. All data are presented as mean ± SEM. Number of experiment repeats, biological replicates and p values are indicated in figure legends.

## Acknowledgements

This work was supported by the National Institutes of Health (HL115141, HL134849, HL148207, HL148355, HL153356 to MWF) the Arthur K Watson Charitable Trust (to MWF), the Dr. Ralph and Marian Falk Medical Research Trust (to MWF), the Swiss National Science Foundation (P2BEP3_162063 to SH), American Heart Association (18POST34030395 to SH.; and 18SFRN33900144 and 20SFRN35200163 to MWF) and National Natural Science Foundation of China (81570334 and 81770358 to TY) and Xiangya Eminent Doctor Project (#013 TY). We thank Zhiyong Deng for histology technical assistance.

## Additional information

### Funding

| Funder | Grant reference number | Author |
|---|---|---|
| National Institutes of Health | HL115141 | Mark W Feinberg |
| National Institutes of Health | HL134849 | Mark W Feinberg |
| American Heart Association | 18SFRN33900144 | Mark W Feinberg |
| American Heart Association | 18POST34030395 | Stefan Haemmig |
| Falk Foundation | | Mark W Feinberg |

| | | |
|---|---|---|
| National Natural Science Foundation of China | | Tianlun Yang |
| National Institutes of Health | HL148207 | Mark W Feinberg |
| National Institutes of Health | HL148355 | Mark W Feinberg |
| National Institutes of Health | HL153356 | Mark W Feinberg |
| Arthur K Watson Charitable Trust | | Mark W Feinberg |
| Dr. Ralph and Marian Falk Medical Research Trust | | Mark W Feinberg |
| Swiss National Science Foundation | P2BEP3_162063 | Stefan Haemmig |
| American Heart Association | 18POST34030395 | Stefan Haemmig |
| American Heart Association | 18SFRN33900144 | Mark W Feinberg |
| American Heart Association | 20SFRN35200163 | Mark W Feinberg |
| National Science Foundation | 81570334 | Tianlun Yang |
| National Science Foundation | 81770358 | Tianlun Yang |
| Xiangya Hospital | #013 | Tianlun Yang |

The funders had no role in study design, data collection and interpretation, or the decision to submit the work for publication.

## Author contributions

Dafeng Yang, Data curation, Formal analysis, Investigation, Visualization, Methodology, Writing - original draft; Stefan Haemmig, Data curation, Formal analysis, Investigation, Visualization, Methodology, Writing - review and editing; Haoyang Zhou, Daniel Pérez-Cremades, Data curation, Formal analysis, Investigation, Methodology; Xinghui Sun, Investigation, Methodology, Writing - review and editing; Lei Chen, Data curation, Investigation; Jie Li, Investigation, Methodology; Jorge Haneo-Mejia, Resources, Writing - review and editing; Tianlun Yang, Resources; Ivana Hollan, Resources, Investigation; Mark W Feinberg, Conceptualization, Resources, Data curation, Formal analysis, Supervision, Funding acquisition, Validation, Investigation, Visualization, Methodology, Writing - original draft, Project administration, Writing - review and editing

## Author ORCIDs

Mark W Feinberg https://orcid.org/0000-0001-9523-3859

## Ethics

Animal experimentation: All mice were maintained under SPF conditions at an American Association for the Accreditation of Laboratory Animal Care-accredited animal facility at the Brigham and Women's Hospital (protocol #2016N000182). All animal protocols were approved by the Institutional Animal Care and Use Committee at Harvard Medical School, Boston, MA and conducted in accordance with the National Institutes of Health Guide for Care and Use of Laboratory Animals.

## Decision letter and Author response

Decision letter https://doi.org/10.7554/eLife.58064.sa1
Author response https://doi.org/10.7554/eLife.58064.sa2

# Additional files

## Supplementary files

• Supplementary file 1. *MiR181a2b2* promoter sequences.
• Supplementary file 2. Primer list.
• Transparent reporting form

## Data availability

Source data files have been provided for Figures 1 and 2. RNA-Seq data has been made accessible.

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
