## [Decision Letter]

**Acceptance summary:**

This paper outlines a microRNA-dependent mechanism for the anti-inflammatory actions the widely used drug methotrexate. The authors responded constructively to the majority of the concerns raised during review and the new data added has strengthened the conclusions.

**Decision letter after peer review:**

Thank you for submitting your article "Methotrexate attenuates vascular inflammation through an adenosine-microRNA dependent pathway" for consideration by *eLife*. Your article has been reviewed by two peer reviewers, one of whom is a member of our Board of Reviewing Editors, and the evaluation has been overseen by Matthias Barton as the Senior Editor. The reviewers have opted to remain anonymous.

The reviewers have discussed the reviews with one another and the Reviewing Editor has drafted this decision to help you prepare a revised submission.

Summary:

Yang et al. describe a mechanism by which MTX treatment reverts insulin resistance and adipose inflammation. The authors report that the MTX metabolite adenosine, acting through Adenosine receptor 3, induces the expression of miR181b2, a microRNA previously shown by the same group to stabilize AKT and inhibit NF-κB signaling. The authors utilized endothelial cell-specific deletion of miR181a2/b2 to demonstrate that the effects of MTX on reducing adipose macrophage accumulation and insulin resistance rely on miRNA181b. Overall the finding are of interest but there are several concerns regarding methodology and experimental interpretation that would need to be addressed in a revised manuscript.

Essential revisions:

1) The magnitude of the induction of miR-181 by MTX or adenosine is small, with the majority of the changes in miR-181 ranging in less than a 1.5-fold increase in expression upon methotrexate treatment. While acknowledging that it is difficult to ascertain how many copies of a given miRNA are necessary to exert a functional effect, miR-181 is abundantly expressed in endothelial cells, therefore it is difficult to imagine that a less than 50% increase in expression would exert a meaningful change in functionality. This issue should be addressed and discussed.

2) Related to point 1, as the remainder of the manuscript suggests that miR181b is required for effects of MTX, is it possible that MTX-adenosine regulates the downstream function of miR181b (e.g. miRNA processing, regulate miRNP formation/miRNA binding protein abundance, etc..) instead of acting solely through transcriptional upregulation?

3) The concentrations of adenosine used in vitro in Figure 1 are very high compared to plasma levels of adenosine in humans. The authors use a dose range from 1-100uM, and in published work in humans the highest level measured was 2uM (see PMID:31379241). The authors see no effect of adenosine on miR-181 expression or other measurements at this concentration. It is difficult to conclude that adenosine is the mechanism by which MTX exerts its effects.

4) In order to conclude that the effects of MTX in ECs is mediated through miR-181, the authors need to do a more in-depth analysis of EC inflammation beyond VCAM1 expression. This is also important for the link in vivo in the high-fat diet feeding model in Figures 4 and 5. The characterization of EC inflammation via VCAM1 expression isn't sufficient to conclude that MTX alters insulin sensitivity via miR-181.

5) The mechanistic connection between SMAD3/4, miR-181 and the effects of methotrexate on insulin sensitivity need to be better defined. How do the authors believe that SMAD3/4 induction of miR-181 by MTX is occurring with respect to insulin sensitivity? While the promoter data presented in Figure 3 are convincing, it does not link well to the in vivo data presented in Figures 4-5.

6) The authors should explain why vehicle and 4-OHT injected experimental groups, instead of an experimental set up with cre-/cre+ and 4-OHT injection to both genotypes, were used in the experiments in Figure 5. The current scheme does not exclude the effect of 4-OHT on the readout. Tamoxifen has been reported to affect glucose metabolism (Ceasrine et al., 2019).

---

## [Author Response]

Essential revisions:1) The magnitude of the induction of miR-181 by MTX or adenosine is small, with the majority of the changes in miR-181 ranging in less than a 1.5-fold increase in expression upon methotrexate treatment. While acknowledging that it is difficult to ascertain how many copies of a given miRNA are necessary to exert a functional effect, miR-181 is abundantly expressed in endothelial cells, therefore it is difficult to imagine that a less than 50% increase in expression would exert a meaningful change in functionality. This issue should be addressed and discussed.

We thank the reviewer for this point. We and others have demonstrated that moderate changes in miRNA expression (up or down) can have profound functional impact in the vascular endothelium in response to divergent acute and chronic inflammatory conditions (e.g. atherosclerosis, sepsis, thrombosis, and insulin resistance, among others) (Guo et al., 2018; Lin et al., 2016; Ma et al., 2016; Sun et al., 2014; Sun et al., 2012; Sun et al., 2016). We have incorporated this point in the Discussion section and revised in the text:

“MiRNAs are key post-transcriptional gene regulators and even moderate changes in miRNA expression can have profound functional impact in the vascular endothelium in response to divergent acute and chronic inflammatory conditions. […] Overexpression of miR-181b by tail vein injection of miR-181b mimics (~1.5-2.5-fold increase in miR-181b expression) improved acute vascular inflammation induced by endotoxemia (Sun et al., 2012), and chronic vascular inflammation triggered by atherosclerotic plaque formation (Sun et al., 2014), obesity and insulin resistance (Sun et al., 2016), and thrombin-induced arterial thrombosis (Lin et al., 2016).”

2) Related to point 1, as the remainder of the manuscript suggests that miR181b is required for effects of MTX, is it possible that MTX-adenosine regulates the downstream function of miR181b (e.g. miRNA processing, regulate miRNP formation/miRNA binding protein abundance, etc..) instead of acting solely through transcriptional upregulation?

While we find the MTX-adenosine impacts the pri-miR-181b and the promoter (transcriptionally), we cannot rule out the possibility that non-transcriptional mechanisms may be involved. We have incorporated this interesting point in the Discussion section. We have added new data in Figure 3—figure supplement 2 and incorporated this interesting point in the Discussion section and revised the text:

“At the post-transcriptional level, mRNA stability is an important factor determining mRNA abundance. […] These results indicate that the promoter region between -402 and -301 is likely required for MTX and Ad-induced transcriptional activity of the miR-181a2b2 gene and MTX and Ad may affect pri-miR-181b2 expression at both transcriptional and post-transcriptional levels.”

“Mechanistically, MTX activated an Ad-ADORA3-SMAD3/4 signaling cascade that transcriptionally activated miR-181a2b2 expression and modestly increased pri-miR-181b2 mRNA stability at the post-transcriptional level.”

3) The concentrations of adenosine used in vitro in Figure 1 are very high compared to plasma levels of adenosine in humans. The authors use a dose range from 1-100uM, and in published work in humans the highest level measured was 2uM (see PMID:31379241). The authors see no effect of adenosine on miR-181 expression or other measurements at this concentration. It is difficult to conclude that adenosine is the mechanism by which MTX exerts its effects.

We thank the reviewer for this point. It is generally thought that the concentrations of extracellular adenosine are below 1uM in unstressed tissues (PMID:11008982), whereas adenosine levels in inflamed or ischemic tissues can be as high as 100uM (PMID:14698282).

For example, plasma adenosine levels reached 4-10uM in patients with sepsis and in the 10-100uM range in the synovial fluid of patients with rheumatoid arthritis (PMID:11008982, 11248511). Moreover, similar results were found by a recent published work: 100uM adenosine significantly repress TNF-a-induced multiple adhesion molecules (VCAM-1, ICAM-1 and Eselectin) (PMID: 29038540). Consistent with these reports, we found that adenosine at 1uM, simulating “basal non-pathological” conditions did not affect miR-181b expression, whereas treatment with MTX or higher concentrations of adenosine (5-100uM) increased miR-181b expression (Figure 1B and E). We also find an increased trend in miR-181b expression with 5uM adenosine (Figure 1E). We have incorporated this point in the Discussion section and revised the text:

“Ad is continuously released by cells to the extracellular environment at a low concertation (less than 1 µM) in unstressed tissues, whereas Ad concentration can reach as high as 100 µM in inflamed or ischemic tissues (Hasko and Cronstein, 2004; Jacobson and Gao, 2006). […] Moreover, we demonstrated that the MTX and Ad-mediated protective effects on ECs were dependent on miR-181b expression.”

4) In order to conclude that the effects of MTX in ECs is mediated through miR-181, the authors need to do a more in-depth analysis of EC inflammation beyond VCAM1 expression. This is also important for the link in vivo in the high-fat diet feeding model in Figures 4 and 5. The characterization of EC inflammation via VCAM1 expression isn't sufficient to conclude that MTX alters insulin sensitivity via miR-181.

We thank the reviewer for this point. We have now performed new RNA-Seq studies: (1) in eWAT from the systemic miR-181a2b2 KO and flox control mice (i.e. under HFD) (Figure 4—figure supplement 2A-B); and (2) in eWAT from the systemic miR-181a2b2 KO and flox control mice in the presence of MTX (i.e. Under HFD+MTX) (Figure 4—figure supplement 2C-D) to obtain a more comprehensive portrait of the transcriptomic changes in response to high-fat diet and MTX. Consistent with our in vitro and in vivo findings for VCAM-1, Gene Set Enrichment Analyses (GSEA) of the mice under HFD conditions revealed the enrichment for several top pathways for cell adhesion (Figure 4—figure supplement 2A). Further exploration into the network analyses for these cell adhesion pathways demonstrated multiple adhesion molecules (VCAM-1, ICAM-1, PECAM-1, others) that were also regulated (Figure 4—figure supplement 2B). Interestingly, GSEA of the mice under HFD + MTX conditions revealed additional enrichment pathways related to cell cycle (Figure 4—figure supplement 2C-D). These more comprehensive analyses are now incorporated into the Results and Discussion sections:

“To better characterize the underlying mechanisms how MTX and miR-181a2b2 affects insulin resistance, we performed next-generation sequencing of RNA isolated from eWAT of these HFD-fed miR-181a2b2 flox and KO mice with or without MTX treatment. […] In eWAT from MTX-treated miR-181a2b2 flox and KO mice, GSEA found that cell cycle, inflammation, and chemotaxis pathways were more affected by MTX (Figure 4—figure supplement 2C-D).”

“Moreover, our data suggest that high-dose of MTX (1mg/kg) is essential to increase miR-181b expression and modulate a large gene network involving EC-leukocyte adhesion, chemotaxis, immune response, and cell cycle (Figure 4—figure supplement 2) in eWAT; consequently, this may ameliorate EC activation, insulin resistance, and adipose tissue inflammation. Several of these identified pathways (EC-leukocyte adhesion, chemotaxis, and immune response) are consistent with pathways known to be regulated by miR-181b (Guo et al., 2018; Lin et al., 2016; Ma et al., 2016; Sun, He, et al., 2014; Sun et al., 2012; Sun et al., 2016).”

5) The mechanistic connection between SMAD3/4, miR-181 and the effects of methotrexate on insulin sensitivity need to be better defined. How do the authors believe that SMAD3/4 induction of miR-181 by MTX is occurring with respect to insulin sensitivity? While the promoter data presented in Figure 3 are convincing, it does not link well to the in vivo data presented in Figures 4-5.

To better link our mechanistic studies in Figure 3 with the in vivo studies, we have performed additional studies that assessed the protein expression by Western of Smad2, Smad3, and Smad4 in eWAT from EC-miR-181a2b2 KO and flox control mice under HFD and HFD + MTX conditions. These data (Figure 5—figure supplement 3) demonstrate that Smad2 and Smad3 are markedly reduced in eWAT in the EC--miR-181a2b2 KO mice under HFD + MTX conditions compared to flox-cre controls, but not without MTX treatment. Given the role of miR-181b in mediating effects from MTX/Smad signaling, these findings suggest that there may be a positive feedback mechanism by which the loss of miR-181b attenuates Smad expression in response to MTX. We have incorporated these findings in the Results and Discussion sections:

“Because we identified that SMAD3/4 signaling contributed largely to miR-181b2 expression (Figure 3), we next investigated expression of SMADs in eWAT tissues from these HFD-fed flox-Cre and iEC-KO mice. We found that SMAD2 and SMAD3 expression decreased in MTX-treated iEC-KO mice compared to flox-Cre mice, whereas there were no differences between flox-Cre and iEC-KO mice without MTX treatment, suggesting a feedback mechanism involved in the MTX-SMAD-miR-181b cascade (Figure 5—figure supplement 3).”

“We also found that SMAD2 and SMAD3 expression decreased in eWAT of MTX-treated iECKO mice compared to flox-Cre mice only in response to MTX treatment. Future studies will need to clarify the mechanisms underlying a possible positive feedback loop involving the MTX-SMAD-miR-181b signaling pathway.”

6) The authors should explain why vehicle and 4-OHT injected experimental groups, instead of an experimental set up with cre-/cre+ and 4-OHT injection to both genotypes, were used in the experiments in Figure 5. The current scheme does not exclude the effect of 4-OHT on the readout. Tamoxifen has been reported to affect glucose metabolism (Ceasrine et al., 2019).

We thank the reviewer for this point. However, the study by Ceasrine et al., 2019, explored the role of tamoxifen on GTT after early time points of 1-3 weeks after injection. In addition, they found that oral tamoxifen delivery did not affect the GTT and resulted in lower plasma tamoxifen levels, but higher circulating levels of tamoxifen metabolites such as 4-hydroxytamoxifen (4-OHT), which is what we are using in this study. Furthermore, our GTT/ITT studies were performed almost 12 weeks after initial injection making the likelihood of impact from 4-OHT extremely low. Finally, numerous studies demonstrate similar experimental control as we did for 4-OHT (Payne, De Val and Neal, 2018; Zhang et al., 2016)(Liu et al., 2015; Tian et al., 2013). Nonetheless, we have incorporated this point as a potential limitation in the Discussion section and revised the text:

“Finally, while tamoxifen has been reported to affect GTT at early time points of 1-3 weeks after injection (Ceasrine et al., 2019), in the current study we performed GTT and ITT at 12 weeks after initial injection of the tamoxifen metabolite 4-hydroxytamoxifen, making the likelihood of impact from 4-OHT extremely low as reported (Payne, De Val and Neal, 2018; Zhang et al., 2016) (Liu et al., 2015; Tian et al., 2013).”